# Determinants of healthful eating and physical activity among adolescents and young adults with type 1 diabetes in Qatar: A qualitative study

Hanan AlBurno[1]*, Liesbeth Mercken[1,2], Hein de Vries[1], Dabia Al Mohannadi[3], Francine Schneider[1]

1 Care and Public Health Research Institute (CAPHRI), Maastricht University, Netherlands, The Netherlands, 2 Faculty of Psychology, Department of Health Psychology, Open University of The Netherlands, Heerlen, The Netherlands, 3 Department of Endocrinology and Diabetes, Hamad General Hospital, Doha, Qatar

* H.Burno@maastrichtuniversity.nl

## Abstract

### Background

In Qatar, as in the rest of the world, the sharp rise in the prevalence of type 1 diabetes (T1D) is a leading cause for concern, in terms associated with morbidity, mortality, and increasing health costs. Besides adhering to medication, the outcome of diabetes management is also dependent on patient adherence to the variable self-care behaviors including healthful eating (HE) and physical activity (PA). Yet, dietary intake and PA in adolescents and young adults (AYAs) with T1D are known to fall short of recommended guidelines. The aim of this study was to develop an in-depth understanding of the behavioral determinants of HE and PA adherence among Arab AYAs within the age range of 17–24 years with T1D attending Hamad General Hospital.

### Methods

Semi-structured, face-to-face individual interviews were conducted with 20 participants. Interviews were based on an integrative health behavior change model, the I-Change model (ICM). All interviews were audio-recorded, transcribed verbatim, and analyzed using the framework method.

### Results

More participants reported non-adherence than adherence. Several motivational determinants of adherence to HE and PA were identified. The majority of participants were cognizant of their own behaviors towards HE and PA. Yet, some did not link low adherence to HE and PA with increased risks of health problems resulting from T1D. Facilitators to adherence were identified as being convinced of the advantages of HE and PA, having support and high self-efficacy, a high level of intention, and a good health care system.

**Data Availability Statement:** All relevant data are within the article.

**Funding:** This study was supported by a grant from Hamad Medical Cooperation (# 17017/17). https://site.hamad.qa/EN/Pages/default.html. Funders had no role in study design, data collection and analysis, decision to publish, or preparation of the manuscript.

**Competing interests:** The authors have declared that no competing interests exist.

## Conclusion

The suboptimal adherence in AYAs to HE and PA needs more attention. Supportive actions are needed to encourage adherence to a healthy lifestyle to achieve benefits in terms of glycemic control and overall health outcomes, with a special focus on adolescents. Interventions are needed to foster motivation by addressing the relevant determinants in order to promote adherence to these two behaviors in AYAs with T1D.

## Introduction

Type 1 diabetes (T1D) is the major type of diabetes in adolescents and young adults (AYAs) [1,2] accounting for more than 85% of all diabetes cases in AYAs under the age of 20 worldwide [1]. Globally, the prevalence of T1D in children and young adults has doubled since the nineteen nineties and is expected to double again in the coming few years [3] at an annual increment rate of 2–5% in many countries [4,5]. The Arab world has one of the highest global incidence and prevalence rates of T1D [6–8]. In Qatar, as in the rest of the world, the sharp rise in the prevalence of T1D is a leading cause for concern, in terms associated with morbidity, mortality, and increasing health costs [9,10]. AYAs with T1D are at an increased risk of developing diabetes-related complications including retinopathy, nephropathy, neuropathy, and cardiovascular disease at an early age [11,12].

T1D is a chronic disease that needs to be largely self-managed. Hence, the outcome of diabetes management is highly dependent on patient adherence to personal self-care behaviors, including healthful eating (HE) and physical activity (PA) [13–15]. However, less attention has been given to adherence to lifestyle in T1D [16,17] than in type 2 diabetes (T2D) [17]. Non-adherence to HE and PA can contribute to poor glycemic control, increased risk of obesity, dyslipidemia, and cardiovascular disease (CVD) in T1D [18–21]. Additionally, a mismatch between carbohydrate intake and insulin can result in hypo- and hyperglycemia, leading to short- and long-term complications [18]. Previous research supports the significance of HE adherence to diabetes outcomes [22–24]. HE has been shown to improve blood glucose control [23] and contribute to weight management in people with T1D [24]. Regular PA, on the other hand, has been shown to reduce cardiovascular risk factors [25,26], decrease insulin resistance, have a beneficial effect on physical fitness [25] and long-term blood glucose control [27].

Given the importance of HE and PA for health outcomes for people living with diabetes, international guidelines promote its adherence in this population. The American Diabetes Association (ADA) guidelines recommend that "adolescents with T1D should be advised to perform at least 60 min per day of moderate- or vigorous-intensity aerobic activity, with strength training at least 3 days per week" [28,29]. Additionally, most young adults with T1D should be advised to perform at least 150 min/week of moderate-intensity aerobic PA (50–70% of maximum heart rate). Patients should also be encouraged to perform resistance exercise "at least two times per week on non-consecutive days" [29,30]. AYAs with T1D should also be encouraged to make improved food choices, limit consumption of fat-containing foods, added sugar and sodium, and consume sufficient amounts of whole-grain foods, vegetables, and fruits [18,31]. They also need to match insulin dose to carbohydrate intake. National and international guidelines call for dietetic advice on HE and carbohydrates counting techniques to be part of the routine diabetes review [29,32,33]. Further, the ADA recommends that all AYAs with T1D should limit energy consumption from saturated fat to less than 7% [34]. Research has highlighted the importance of individual dietary advice for people with T1D

[35]. A higher level of dietetic input is certainly appropriate for people with living T1D on intensive insulin therapy regimes [36].

Despite these recommendations and known benefits of adherence, HE and PA in AYAs with T1D are known to fall short of recommended guidelines [15,18,23,37,38]. The published estimates of non-adherence rates to HE behaviors have ranged from 21% to 95% across studies [18]. It was found that adolescents with T1D were less active compared to their peers [37]. According to some studies, people living with diabetes have more difficulties adhering to a suitable diet and PA than to insulin medication [39,40]. These findings affirm the need for further investigations into adherence to HE and PA and their corresponding determinants in AYAs with T1D to better understand non-adherence and to guide developing interventions to increase adherence.

Several behavioral determinants have been found to affect patient adherence and, hence, metabolic control. These are related to various individual, social, and environmental variables [18,41]. At the individual level, research has identified knowledge, outcome expectations, emotional factors, self-efficacy, motivation, physical skills, and goal setting as important factors to self-care behaviors including HE and PA [42–45]. At the social and environmental levels, communication with health care providers, and cultural, social, and family support were identified as important factors [46,47]. Still, suboptimal adherence and gaps in the literature exist. Additionally, many studies do not include all relevant determinants, hindering a comprehensive overview of its importance and decisions concerning what to focus on in interventions. Very few studies are available in Qatar and the rest of the Arab world that can be used to guide intervention development for AYAs. Consequently, this study employed an integrative model, the I-Change model (ICM) [48–50] to facilitate a more comprehensive understanding of these two lifestyle behaviors and their corresponding determinants in people living with T1D. The ICM has been successfully used to predict and change lifestyle behaviors in people without diabetes [49–53] and in people living with type 2 diabetes [54,55]. An essential feature of the ICM is that it acknowledges three phases in the process of behavior change: awareness, motivation, and action. These phases are particularly relevant for diabetes control as several patients are not optimally aware of non-adherence to the recommendations and do not plan or execute relevant actions to realize the prescribed recommendations. The ICM suggests that motivation factors (attitude, social influence beliefs, and self-efficacy) are influenced by different pre-motivational factors such as awareness factors (cognizance, knowledge, risk perceptions, and cues to action) and predisposing factors such as information factors (the quality of messages, channels, and sources used) [48,50]. Within the post-motivational phase, factors that increase the likelihood of intention being translated into action are action and coping planning [48]. Additionally, it is important to explore determinants within the dynamics of different family, social, and cultural environments. A greater understanding of health issues related to culture is critical because cultural beliefs and practices may facilitate or discourage diabetes management. In Qatar, factors related to HE and PA were reported in adults [56–58] and adolescents from the general population [59], women with coronary disease [60], and in children with T1D [61]. Results showed the specific socio-cultural context of this region influences the decisions and behaviors to participate in a healthy lifestyle. However, evidence from AYAs with T1D is lacking. In this context, understanding the behaviors, their social-ecological and motivational determinants of individuals with T1D is vital to meet the complex demands of managing diabetes [62,63]. To the best of our knowledge, no research has studied this in Qatar. In sum then, the objectives of this study were, first, to identify the behaviors of AYAs with T1D towards HE and PA and, second, to examine the facilitating and hindering factors associated with these behaviors.

## Method

### Research design

This study used a qualitative design by means of semi-structured, face-to-face individual interviews. Qualitative description design, frequently used in health behavior research, is recommended to describe as well as understand participants' perspectives on a phenomenon [64,65]. This is essential to facilitate engaging diverse participants' perspectives, integrating findings into the design and conduct of future research [66], and into more effective diabetes management programs and services aimed at minimizing barriers and maintaining/promoting facilitators towards HE and PA. Individual interviews were chosen to make allowances for cultural barriers on disclosing sensitive issues, but also to encourage open discussion in a free way without reservation or group factors, as research has shown that young people are often reserved about expressing themselves in front of other people [67]. The Institutional Review Board (IRB), Medical Research Centre Committee-Hamad Medical Corporation granted ethical approval. Research number 17017/17. Data was collected over a period of four months (October 2017 –January 2018).

### Participants and recruitment

The target population of interest in this study consisted of AYAs living with T1D within the age range of 17–24 years, Qatari and non-Qatari (from countries in the Gulf Cooperation Council (GCC) and the Middle East and North Africa (MENA) region) living in Qatar, and who had been diagnosed with T1D for at least two months prior to the study. This was based on evidence from behavior psychology research which estimates the average time required for a new behavior to become automatic (assuming that the patients will start the new healthy behaviors at the time of their diagnosis, although some may be already engaged in HE and PA before their diagnosis) to be around 66 days [68]. The selection of an all-Arab sample was aimed at providing a genetically and/or culturally near-homogenous sample population to help eliminate extraneous factors that could act as potential confounders, such as socio-cultural factors like food habits. Patients with secondary diabetes that is a consequence of another medical condition or consumption of certain medications such as corticosteroids, and people with cognitive impairments, drug or alcohol dependence, or gestational diabetes were excluded.

In order to recruit AYAs, a purposive sampling method was used [69,70], i.e. physicians attending patients intentionally selected participants who met the predefined eligible criteria and referred them to the principal investigator (PI) (first author) for confirmation of eligibility. This sampling method was used to recruit patients with a diverse range of metabolic control (optimal, suboptimal, and poor), aiming to broaden the perspective on the topic. In adolescents, optimal metabolic control was defined as having HbA1c value <7.5% (<58mmol/mol), a poor metabolic control denoted an HbA1c >9% (>75mmol/mol), and suboptimal metabolic control was when the HbA1c value lay between 7.5–9% (58-75mmol/mol) [71]. In young adults, HbA1c levels were categorized into <7% (<53mmol/mol), 7–7.9% (53-63mmol/mol), and ≥8%(≥ 64mmol/mol), reflecting optimal, suboptimal, and poor metabolic control, respectively. This was based on the international HbA1c consensus committee recommendations [72] and clinical practice guidelines' recommendations [73–75]. At the beginning of each interview, the PI obtained the informed written consent forms from young adults aged 18 years or older and the written assent forms from those under 18 years old with their carers' consents, using the procedure approved by the ethics committee. The forms included a sufficient explanation of the study.

## Sample size

A total of 20 interviews was judged to be suitable for determining sample size adequacy to achieve data saturation based on similar previous research [76,77]. Additionally, we were guided by saturation parameters found in prior studies, i.e. focused research questions [78,79], mainly theory-driven themes [80], a relatively homogeneous sample, relatively long individual interviews, and the use of an intensive framework analysis strategy [81]. Further, we examined the depth and the richness of the collected information by using an analysis saturation grid during the analysis process [82].

## Interview process and procedure

The first author (HB), trained in patients' consultation and qualitative research, interviewed the participants using an interview guide. HB did not have a previous relationship with the participants. The interviews were conducted in a quiet area at the diabetes clinics in Hamad General Hospital, which is the major public hospital in Doha, Qatar, and were scheduled for approximately 60 minutes. Participants were assured that all data would be anonymized. At the beginning of each interview, the participants were familiarized with their diabetes care team and ADA recommendations for HE and PA [34,83] and were asked about demographics and clinical characteristics such as age, level of education, duration of diagnosis with diabetes, etc. All interviews were audio-recorded and transcribed verbatim.

## Tool development

An open-ended interview guide was piloted and culturally adapted prior to its use. The guide was developed based on the socio-cognitive constructs in the ICM proposed by Vries [48] (Table 1). The guide was reviewed by an advisory group consisting of experts (e.g., those with

**Table 1. Interview guide.**

| Topic | Discussion |
|---|---|
| **Pre-motivational section** | |
| Awareness factors | awareness of one's behavior (cognizance): asking young people about their adherence to healthful eating (HE) habits and performing physical activity (PA), based on the agreed recommendations from their diabetes care team. |
| | awareness of the level of diabetes control (cognizance): whether their treatment regimen is controlled and why. |
| | awareness of the need to change (cognizance). |
| Risk perceptions | perceived susceptibility and severity of diabetes complications. |
| **Motivations or intentions section** | |
| Individual's attitude | identification of advantages and disadvantages of behaviors related to being active and eating healthy food. |
| Social influences | participants' recognition of the support that they encounter from others in carrying out the behavior. |
| Self-efficacy | situations in which a person finds it easy/difficult to eat healthy food/perform physical activity. |
| **Post-motivational section: Action planning** | |
| Preparatory planning | plans to help the participant to undertake attempts towards performing physical activity and eating healthy food. |
| Coping or maintenance planning or | plans how to cope with difficult situations, barriers, and relapse. |
| **Distal predisposing factors** | |
| Information factors | related to the quality of messages, channels and sources used. |

knowledge and experience) in the areas of change theories, qualitative research, and T1D specialized health care providers (HCPs). Changes were made as appropriate, based on both a field test and expert opinion. The guide was aimed at enabling the interviewer to identify behavioral determinants while at the same time allowing some more exploration by using prompts and probes. The socio-demographic and medical background included information on gender, age, educational level, and diabetes and insulin history.

## Data analysis

### Demographic and clinical characteristics

Results of demographic and clinical characteristics data were expressed as mean [standard deviation (SD)] or percentage of total responses. Adherence and non-adherence were self-reported and assessed in the interview. At the beginning of the interviews, the participants were familiarized with their diabetes care team and ADA recommendations for HE and PA. Then, based on their answers during the interview, they were considered adherents; those respondents who reported that they always or most of the time follow the agreed recommendations from their diabetes care team for HE and PA; otherwise they were considered non-adherents. The percentages of patients with optimal, suboptimal, or poor metabolic control were determined as reflected by HbA1c, as an index of glycemic control over the previous 6–8 weeks. Data on HbA1c was collected from patients' records, the purpose was to compare HbA1c results with participants' perceptions of their level of control, and whether they perceive diabetes adherence to HE and PA would influence diabetes outcome.

### Qualitative data analysis

The interviews were audio-recorded and transcribed verbatim in Arabic before being translated into English. For data management and analysis, the framework method was used, which involves a combination of inductive and deductive approaches [84–87]. This method was deemed appropriate because of the need to both describe and interpret the diabetes self-management behavioral predictors. This method consisted of the following steps: (1) familiarization with the interview; during this step, the interviewer (first author) read the transcripts several times to produce an overall, general impression of the data; (2) coding; during this process, HB coded all transcripts by allocating text segments to multiple codes to account for the complexity of data and double-checked against the codes. FS and LM reviewed the codes to enhance the quality of the coding process [65,88–90]. Then, codes and sub-codes were grouped to form main themes and categories. A codebook was developed using predefined themes from the interview guide; additional codes and sub-codes that emerged during data analysis were added to the initial codebook [91,92]; (3) developing the analytical thematic framework; a coding tree was created to form the working analytical framework; (4) indexing; applying the analytical framework, during which the thematic framework was applied to all transcripts and supplemented with new emerging themes and categories; (5) charting data into the framework matrix, where participants' responses were summarized in a matrix for each health behavior; and (6) mapping and interpretation. To promote the reliability and validity of data during coding and analysis processes, thus rigor, data verification strategies were used [93]. This was achieved by (1) employing inductive and deductive methods, the researchers HB, LM, and FS regularly and iteratively discussed the coding system and analysis to validate the consistency in the application of codes, data interpretation, and formulation of findings [93,94]; (2) using a common conceptual framework with a priory defined codes, which were specific to particular interview questions [65,89]; (3) performing concurrent data collection and analysis to ensure methodological coherence; and (4) checking for sampling

adequacy and saturation. Moreover, in order to maintain consistency with the study aim, we applied an "ad hoc unitization strategy" by including theoretically relevant simultaneous and interpretative codes in the coding frame and in the analysis and interpretation of data [65].

## Results

### Demographic characteristics and medical status information

Out of the 20 interviewees, 55% were Qatari, and the distribution of males and females was equal. The mean age was 21.6 years (SD = 2.6). Demographic and medical status information are shown in Table 2.

### Pre-motivational factors

**Awareness.** Participants discussed topics related to their awareness of their behavior; awareness of risk perception of their own behavior; how these behaviors impacted their level

**Table 2. Main characteristics of the sample (n = 20).**

| Characteristic | Number (%) |
|---|---|
| **Adherence** | |
| Adherents | 5 (25) |
| Non-adherents | 15 (75) |
| **Gender** | |
| Male | 10 (50) |
| Females | 10 (50) |
| **Age, years** | |
| ≥17 - <18 (adolescents) | 7 (35) |
| ≥18–24 (young adults) | 13 (65) |
| **Nationality** | |
| Qatari | 11 (55) |
| Other Gulf Cooperation Council (GCC) countries | 2 (10) |
| Other Arab countries | 7 (35) |
| **Education level** | |
| Secondary | 7 (35) |
| Graduate & above | 13 (65) |
| **Duration of diabetes** | |
| 1–5 year | 3 (15) |
| 6–10 years | 3 (15) |
| >10 years | 14 (70) |
| **Evidence of late diabetes complications** | |
| Yes | 2 (10) |
| No | 18 (90) |
| **If yes, which** | |
| Kidney | 1 (5) |
| Eyes | 1(5) |
| **insulin administration device** | |
| Injectable pen | 9 (45) |
| Insulin Pump | 11 (55) |
| **HbA1c Category across all age groups** | |
| Optimal Metabolic Control | 2 (10) |
| Suboptimal Metabolic Control | 6 (30) |
| Poor Metabolic Control | 12 (60) |

of diabetes control; and awareness of the need to change their behaviors, if any. Participants who reported adherence to PA reported being adherent to HE [Quote #1], except for two cases where one patient was adherent to PA but not to HE, and vice versa [Quotes #2 and #3]. In relation to diet, the majority of participants indicated they were not following a dietary plan and that they were aware that they were non-adherent to HE; however, they did mention knowing how food affects their blood sugar levels. Some non-adherent respondents reported not being aware of the need to eat healthier; they remarked that they could eat whatever they wanted as long as they could adjust the insulin dose [Quote #4]. Similarly, many participants reported being aware that they were either inactive or non-adherent to the recommendations for performing PA. Some others indicated that they were already doing some kind of PA, which varied from walking during their working days to going to the gym. Several non-adherent participants realized that their poor HE habits and low level of PA had caused their diabetes to be uncontrolled [Quote #5]. Other reasons provided for poor and/or sub-optimal control were irregular sleep, carelessness, and overthinking.

Some respondents overestimated their level of diabetes control and were not fully cognizant of this. They considered themselves to have controlled diabetes, whereas actually, the results of their HbA1c indicated that they were either poorly controlled or had suboptimal control [Quote #6]. When participants were asked about the need to change their behaviors, if any, many non-adherent participants wished to increase activity levels and eat healthier. However, some of them indicated that they were not convinced of changing their behaviors regarding performing PA or HE because they failed to see a relation with diabetes control [Quote #7]. Other non-adherent participants indicated that their daily lifestyles were routines and habits that would be impossible to change [Quote #8].

**Risk perception.**    Most adherent participants and some non-adherent ones recognized the susceptibilities of getting diabetes complications as a result of non-adherence to HE and PA as recommended. They also acknowledged that the risks of complications could be severe. Other non-adherents felt differently and indicated that complications from diabetes would be beyond their control and would happen anyway regardless of what they did, as it is their fate. Some other non-adherents did not link risks of non-adherence to PA, in particular to diabetes complications [Quotes #9 and #10]. Some respondents felt that the complications would happen at an advanced age. A few non-adherent participants thought that the complications would be serious only if they did not administer insulin as recommended or if their HbA1c became high, and thus did not link increased severity of risks to low adherence. One participant did not recall the types of complications of diabetes [Quotes #11 and #12]. Relevant quotes and respondents relating to pre-motivational factors are found in Table 3.

## Motivational factors

**Attitude.**    Irrespective of whether participants were adherents or not, many respondents indicated that HE and PA were linked to health benefits (both physical health and mental/psychological health) and more general advantages. For example, they discussed the advantages of HE and PA on their general diabetes control, such as achieving good glycemic control, and being able to decrease insulin doses [Quote #13]. Some others mentioned preventing, delaying, or avoiding complications as advantages of HE and PA. Some physical health benefits recognized were: losing or maintaining weight, improving body image, fitness, and overall health. In terms of psychological health advantages, respondents mostly mentioned enhancing their mood and living without worries. Additional general benefits for some interviewees were connected to health-related quality of life (HRQL), such as having a healthy life and the possibility of decreasing the chances of developing diseases. Participants also talked about general benefits

**Table 3. Interviewee quotes: Pre-motivational factors.**

| Quote number | Quote and respondent |
|---|---|
| #1 | "I am physically active, I exercise daily. I started one month after being diagnosed with diabetes. I walk for 30–40 min a day, I cycle for another 30 min as well. In gym I lift weights, on weekends, I also swim and play football with my friends. Regarding food, to be honest, at the beginning I didn't like to eat healthy. I used to be very mm.. what do you call "junkie" but since I'm more aware about this I have actually changed my eating habits. For example, I decreased portion size, and I have prevented myself from watching food advertisements. My mother cooks some dishes which are rich in carbs and oil, she would tell me to cook my own food. I cook oats and rice, I cook food with little oil, and I make sure to have salad and healthy food choices." (male, 24, adherent) |
| #2 | "I am not into physical activity, I do not like physical activity, though I eat healthy so as not to gain weight. Three months ago my doctor told me that my weight is above the normal range, since then I have been consistent in eating healthy and I will continue, I have lost almost 7 kg or even more, I want to be in good shape." (female, 19, adherent to healthy eating (HE) not to physical activity (PA). |
| #3 | "No, I don't feel that I need to change anything. I do not feel that I have to eat healthy food because I am physically active." (male, 24, adherent to PA not to HE) |
| #4 | "For me, there is no need to eat healthy, I can eat whatever I want as long as I count my carbs and increase insulin dose, there is no need to stop myself from eating what I want." (female, 18, non-adherent) |
| #5 | "Ok, to be honest, I do not follow my doctor's recommendations and advice, my target goal is to reach A1c of 7 or less if it's possible, I know it's my mistake, I need to make healthy food choices and to do physical activity." (female, 24, non-adherent) |
| #6 | "My diabetes is controlled, like the last appointment my A1C was around 8.9, so from then I have changed it, what I meant now it is 8.5. Yes, I'm happy with it, as a number I feel it's ok. The most important thing for me is to keep A1C under control to avoid diabetes complications, so 8.5 is Ok with me." (female, 19, non-adherent) |
| #7 | "I do not feel that I have to eat healthy food. I eat anything, especially carbs, but also processed food and such things. When it comes to physical activities, I walk only on weekends for around 45 minutes. Whatever I do, my glucose will always be high. Sometimes I think that I am injecting the wrong dose, or maybe because I am under stress because of school exams and so many things to do." (male, 17, non-adherent) |
| #8 | "What's the point? It's enough for me. I do not force myself. I have to change my lifestyle routine, my diet for diabetes. No–this is enough." (female, 17, non-adherent) |
| #9 | "There is no effect of PA and a healthy diet on diabetes complications, but the glucose will be modified, it will not always be raised, even HbA1C might decrease. My lifestyle will be ok. The complications will occur anyway. I feel that no matter what I do, the complications will still occur." (female, 21, non-adherent) |
| #10 | "Doing PA will not help in reducing complications, not that much, maybe a little. I mean, for example, walking will not reduce the chance of developing diabetes complications. I mean it will help but not that much. I do not feel it has a direct effect on diabetes." (male, 20, non-adherent) |
| #11 | "They [diabetes complications] are not serious, no they are serious for people who actually do not take care of their diabetes, who do not take their insulin as they should, they are serious only if HbA1c results become high." (female, 19, non-adherent) |
| #12 | "I don't know what diabetes complications are. I have forgotten diabetes complications a little bit. It's been a long time since I heard about diabetes complications when I first was diagnosed with diabetes. But I think my blood sugar improves with physical activity so. . . so I exercise sometimes to avoid the complications." (male, 23, non-adherent) |

relating specifically to performing PA [Quote #14]. Concerning healthy eating, the time needed to prepare and make healthy meals was mentioned as the main perceived practical disadvantage for non-adherents. Feeling deprived of their favorite food was the main psychological disadvantage for them. Participants who indicated that they were adherent to HE did not mention any disadvantages. Overall, adherent females reported having concerns about their weight and body image, which was a motivating factor to adhere to HE and PA for them.

Participants who performed PA regularly described physical disadvantages of performing PA, such as muscular pain, an increase in the risk of hypoglycemia and injuries. Non-adherent respondents also mentioned that fear of hypoglycemia (FoH) would hinder their adherence to PA. Time consumption was mentioned as the main practical disadvantage for non-adherents.

Non-adherent participants also mentioned feelings of failure and low self-esteem resulting from their low adherence to PA in comparison with adherent patients [Quote #15].

**Social influence.** Social influence was perceived by the respondents as either positive (prompting the behavior), negative (discouraging), or neutral (no support). The forms of positive social influence the adherents received were emotional and practical, such as exercising with them, cooking healthy food for them, reducing sugary intake, and stopping buying soft drinks [Quotes #16 and #17]. The majority of adherents showed that they mainly got their support from their families, then from professionals, and in a few cases, from observing other adherents. Few participants who did not follow healthy diets or perform regular PA admitted that they would sometimes accept the support and encouragement if given from close friends but not from their families. A few even said that they would listen to the advice about HE and performing PA, but would not follow it [Quote #18].

Friends were mentioned as having a greater negative influence on non-adherent respondents than families and co-workers. Friends mainly encouraged negative behavior, such as encouraging them to either eat sugary and unhealthy foods with them when hanging out or get them to join them instead of going to the gym. They also said that the reasons for their friends' actions were a lack of knowledge about diabetes and the negative consequences of an unhealthy lifestyle on diabetes control. However, this did not stop them from resisting their friends' influence [Quotes #19 and #20]. Some non-adherent participants mentioned that co-workers and families exerted strong pressure to engage in being more active and eating healthier. Yet, this increased their resistance, resulting in doing the opposite [Quote #21].

Some non-adherent participants reported experiencing some forms of diabetes stigma, which had a negative influence on adherence to PA/HE and aggravated the emotional and social impact of diabetes. For instance, adolescents at school reported that some peers avoided interacting with them, thinking that they would not be able to have fun with people living with diabetes because of dietary and PA restrictions, regarding their diabetes as a burden because they had to look after them. A couple of adherents reported restrictions in participating in school activities by some teachers [Quote #22]. Other forms of stigmatization originated from parents and health care practitioners, amplifying a sense of blaming, guilt, and personal failure for not following HCP's advice, regardless of how much they tried [Quote #23]. This negative social influence elicited negative emotions in AYAs such as frustration, anger, and a feeling of being under the control of others. Non-adherent respondents more often mentioned feelings of low self-esteem and social isolation [Quote #24].

**Self-efficacy.** Several situations decreased feelings of self-efficacy in non-adherent participants. The first type of situation mentioned was related to social and cultural customs, prompted by the characteristics of the traditional diet and a lack of support from the environment [Quote #25]. A second type of situation was practical in nature, such as: facing difficulty in preparing healthy meals at home or in counting carbohydrates [Quote #26]. A third type of situation encompassed physical situations: non-adherent participants often mentioned encountering physical difficulties in finding suitable places offering healthy food when eating in restaurants or when traveling [Quote #27]. Easy access to unhealthy food was also indicated as facilitating non-adherence [Quote #28]. A fourth type of situation was personal and psychological. Frequently, participants' erratic lifestyles due to working or studying conditions were mentioned as negatively impacting adherence, as they did not have the time or willpower to find or prepare healthy food. Preferences for unhealthy food accompanied with a dislike for the taste of healthy food were common among them [Quotes #29 and #30].

Concerning PA, a related set of situations were mentioned that often negatively impacted PA adherence. The first type was related to the physical environment. There was a general agreement among non-adherent respondents on the hindering effects of hot weather and

infrastructure conditions, such as the availability of and ease of access to places to walk safely, like sidewalks, walking trails, and so on [Quote #31]. A second type concerned practical and personal situations. Time constraints were viewed as a major personal barrier (e.g., when being occupied with studying, work, or handling life matters). Non-adherent participants mentioned specifically fatigue and lack of sleep due to frequent urination as a result of hyperglycemia, and thus having no energy to engage in PA. Additionally, needing a detailed medical evaluation to determine their fitness to practice PA and the need to undertake additional examinations were viewed as complicated processes and barriers to joining gyms. Therefore, they either did not join or they had to deny that they had diabetes. Having babies/young children at home with no place to leave them during PA was mentioned by some mothers [Quote #32]. A third type was psychological in nature. Non-adherents talked about the overwhelming burden of balancing insulin, diet, and PA before, during, and after PA. A common psychological barrier pertaining to non-adherence to both HE and PA is derived from an intrinsic feeling of restriction, intolerance to routine and commitment, feeling down, boredom, and not seeing results [Quote #33]. FoH was linked specifically to PA.

The situations in which adherents found it easy to eat healthily were associated with social, practical, and psychological situations. For instance, eating with the family and having another family member who has diabetes made adherence easier. Other facilitators to HE were having no restrictions on eating, following an easy diet, and having a day off from the diet program [Quote #34].

The situations in which adherent participants found it easy to perform PA were associated with social, practical, physical, and psychological situations. The social situations were when they practiced with family or when they exercised in gyms because they felt supported by a trained coach and other trainees there. In particular, when the gyms were located near their homes [Quote #35]. Common psychological situations that facilitated adherence to both EH and PA were when they were in a good mood, felt motivated, and when located in their own environment [Quote #36 and #37]. Relevant quotes and respondents relating to motivational factors are found in Table 4.

## Post-motivational factors

**Action planning: Preparatory and coping planning.**   Many non-adherent participants indicated wanting to control their diabetes and to achieve their goals through HE and performing more PA, but lacked self-efficacy and did not set clear goals or make action plans [Quote #38]. However, others mentioned making plans, but failed on plan execution, resulting in their plans failing to be realized [Quote #39]. Non-adherents relapsed more often and found it difficult to resist unhealthy stimuli. The main reasons for relapse provided were boredom, laziness, lack of motivation, and not seeing immediate results [Quotes #40 and #41]. Adherent participants did not make many action plans, but if they did, they were mostly linked to registering in a gym, cooking at home, or decreasing the size of food portions in an attempt to follow a diet [Quote #42].

**Coping planning.**   Non-adherent participants did not make coping plans for the difficult situations they came across when trying to eat healthy; they mentioned relying on increasing the insulin dose to compensate for overeating unhealthy food. Other participants tried to make plans to cut down but not eliminate fast food [Quote #43].

When some adherent participants did make some sort of coping planning, these were mostly connected to plans to decrease portion size and substitute unhealthy food with healthy options, increase their level of PA to compensate for the consumption of extra carbohydrates consumed, and decrease the number of times they went to restaurants [Quote #44].

**Table 4. Interviewee quotes: Motivational factors.**

| Quote number | Quote and respondent |
|---|---|
| #13 | *"Eating healthy and physical activity are good for all people including people living with diabetes. They can maintain my weight and give me happiness hormones. For my diabetes, since I started eating healthy and doing regular physical activity, I managed to decrease insulin doses. I think physical activity and diet might delay the complications of diabetes, they might determine your destiny for the next 50 years."* (female, 23, adherent) |
| #14 | *"When I do PA continuously, I feel relaxed, my blood sugar level becomes normal, it decreases nervousness, my health, in general becomes better, all of that. HE and PA have helped me to live a normal life. I mean that my blood sugar level is controlled; I do not develop such complications and so on. Whenever I do PA and I eat healthy food, I will have a healthy life and I will feel better."* (male, 24, adherent) |
| #15 | *"Performing PA needs time management and makes me feel low-esteemed because other people can achieve this, but not me."* (female, 20, non-adherent) |
| #16 | *"My mother tries to cook food which is suitable for my diabetes and for all members of the family, not making two different meals. my mother hasn't used butter since I was diagnosed with diabetes, and in our home, we don't eat fried food, I think because my mother used to cook this food at home."* (male, 24, adherent) |
| #17 | *"Both parents encourage me to perform physical activity, friends sometimes encourage me to participate in doing physical activity as a group."* (female, 21, adherent) |
| #18 | *"Some of my close friends know that I have diabetes, there is no important reason for all of my friends to know that I have diabetes. Sometimes my close friends will say: don't give her a certain type of food because she has diabetes. Also this might embarrass me. Yes, when they label me as a diseased person, I listen to them, though I don't follow their advice."* (female, 18, non-adherent) |
| #19 | *"All what my friends care about is having fun and hanging out, like: let's go to this café, it offers good cake, this restaurant makes a delicious burger. They stimulate me to eat unhealthy food, but without bad intentions. So, like maybe sometimes they do not think that this unhealthy food is not good for me, you know they don't know much about diabetes. At the end of the day, why can't I do the same, why is it only me? Why do I have to eat differently?"* (female, 18, non-adherent) |
| #20 | *"My friends encourage me to hang out with them instead of going to the gym. They rarely suggest going for exercise. It's not that they are doing it on purpose, because not all of my friends have appropriate and sufficient knowledge about the disease."* (male, 19, non-adherent) |
| #21 | *"I feel tired because I have been doing everything in my power to control my diabetes, but my parents do not seem to appreciate my hard work. My mother and my father, are always like this. I feel sad the most because I get tired, they always keep saying: health, health, take care of your health, take your insulin, go to the gym, do this, do not do that. They kind of force me to do things, this really upsets me and drives me to do the opposite."* (male, 18, non-adherent) |
| #22 | *"My teacher said: 'You. I don't think you are allowed to participate in the match since you have diabetes.' I replied: 'You can't treat me this way. My mom allowed me to do it, this means that she is positive that I can actually control myself.' The teacher says, like: 'OK, you will participate under your own responsibility, if anything happens, we will not be responsible.' I really don't want to be treated like that, we are suffering enough because of diabetes, and on top of that to be treated like this. She talked about my diabetes in front of everybody, though I do not want other students to know that I have diabetes."* (female, 17, adherent) |
| #23 | *"My doctor and my parents treat me like a child. When my tests readings are high, they keep asking questions to make me feel that it is my fault, have you been active? What have you been eating? They keep blaming me for my test results. They might do this because of their concerns, but this really makes me angry and frustrated, I am not a child anymore, I am old enough to take care of myself."* (male, 17, non-adherent) |
| #24 | *"I want to exercise, but I don't have the motivation. It's because of the people around me, they never encourage me, they know I'm at risk of getting complications especially as I grow older, but still there is no support, this really annoys me a lot. That's why I prefer to stay alone most of the time."* (male, 20, non-adherent) |
| #25 | *"Our traditional food contains high amounts of fat and carbohydrates like very sugary tea with milk, rice, sugary coffee. These are our popular kitchen ingredients. It's difficult to avoid such food."* (male, 19, non-adherent) |
| #26 | *"I am not able to prepare healthy meals, because I am single, so I cook for myself. I cook food but salads need to be prepared on the spot. I don't have the energy or desire to prepare it right now. I prepare food for three days, three consecutive days, I feel that my diet could be better. Also, usually I find it difficult to count my carbs in the food I prepare."* (male, 24, non-adherent) |
| #27 | *"When it comes to food it's really hard. I can't hold myself. I eat whatever I want. I can't go to a restaurant and eat just salad. Why would I go there then?"* (male, 23, non-adherent) |

*(Continued)*

**Table 4.** (Continued)

| Quote number | Quote and respondent |
|---|---|
| #28 | *"The supermarket is like steps from my house. I go especially to buy soft drinks. I tried to take lemon juice instead of soft drinks but I couldn't, so I end up taking both of them." (female, 17, non-adherent)* |
| #29 | *"Sometimes when I see videos about people who follow a diet and they lose weight, I get excited to look like them for a while. After that, I get busy with studying, and I feel I'm too lazy to do so. As a result, I end up doing nothing." (female, 18, non-adherent)* |
| #30 | *"Eating healthy is hard. I don't like this kind of food. I don't like fruits and vegetables. I like sweets, I can't resist eating carbs, although it's top of the list of risky food for diabetes." (female, 20, non-adherent)* |
| #31 | *"The weather is one part, another thing is the roads are not suitable for walking. Like if I want to walk from my home to the grocery store that would be very difficult. But (in. . ., where I was studying), you reach your destination in 10 minutes and the country roads are suitable for walking, where you find everyone walking with you. Sometimes, I go to walk in Aspire free walking area, but I need to prepare myself beforehand and that's the difference, where here you need to drive for at least 40 minutes to reach a proper place for walking, but there in . . . it's different. I just go out and walk." (male, 24, non-adherent)* |
| #32 | *"My main problem is being committed to exercise, because I am not allowed to bring my baby with me to the gym." (female, 22, non-adherent)* |
| #33 | *"When I see no improvement, for example, if I have planned to lose weight by doing PA and started to eat healthy, then three months have passed without achieving my target weight I feel like I didn't do anything and I give up." (male, 19, non-adherent)* |
| #34 | *"I used to have one day off in my diet plan so I could eat whatever I want. I can eat in small portions without cutting down what I love to eat." (female, 21, adherent)* |
| #35 | *"Although sometimes I feel that I'm not in the mood for physical activity, I encourage myself to go even when I'm not going to do something, gradually I find myself starting to exercise. So I like to go to the gym because my coach motivates me." (female, 23, adherent)* |
| #36 | *"It depends, if I have the intention and I'm in a good mood to eat healthy, then I will do it, otherwise, it's impossible for me to do it." (female, 17, non-adherent)* |
| #37 | *"Well, when I was in my home country I already adapted to the lifestyle out there. I managed to organize my time, performing PA and eating healthy was a normal thing, it was part of my daily life routine when I was there." (male, 21, non-adherent)* |

Similarly, non-adherent participants indicated not making coping plans for adhering to PA, resulting in performing PA inconsistently, sparsely, or never [Quote #45]. However, adherent participants reported making alternative plans, such as trying to manage time by compensating for days missed for PA; increasing the duration of PA; or doing different kinds of PA, such as walking in the shopping mall or swimming instead of vigorous PA [Quote #46].

## Distal predisposing factors

**Information factors.** The majority of participants, irrespective of whether they were adherents or not, reported that the sources of information available to them were mainly from professionals at diabetes clinics (physicians and diabetes educators) and from the Qatar Diabetes Association. The second common sources were general internet websites, which were in the Arabic language, social media pages, TV, and newspapers [Quote #47]. A couple of adherent participants who used websites specialized in diabetes and traditional methods of media through reading books and journals had followed higher education (university level with medical backgrounds, e.g., pharmacy and nursing).

Some non-adherent participants mentioned friends, mothers, and coaches at gyms as sources of information. Some non-adherent participants looked for diabetes-related information from their friends without diabetes, who in turn used the internet or social media as sources of information. They explained that they did not want to engage in discussions about diabetes with other people living with diabetes, as this would cause them more stress [Quote #48].

Some adherent and non-adherent participants considered the information needed to be updated and modified, because they considered it was complicated, not specific to their situations, and mostly repetitive. They also suggested for the need for more information on what type of exercises are suitable for T1D and on adjusting insulin dose around PA in order to avoid hypoglycemia [Quote #49].

Some participants suggested including PA teachers and other administrative staff in schools in the education about PA and T1D. Others stated that the information related to diabetes and PA was not available to some trainers in the gym. They either did not know how to deal with a hypoglycemic incident, did not have sugary food/drink in the gym, or kept pushing the participant to do more PA without taking their diabetes into consideration [Quote #50]. Relevant quotes and respondents relating to post-motivational and information factors are found in Table 5. The main findings are summarized in Table 6.

**Table 5. Interviewee quotes: Post-motivational and distal information factors.**

| Quote number | Quote and respondent |
| --- | --- |
| #38 | *"I tried to make a plan to control my diabetes, but I couldn't do it. Because it's so hard. I couldn't adapt. Because it's so hard. I don't like plans and I don't like setting goals that are difficult to achieve. If I push myself too hard, I will be the one getting hurt." (female, 20, non-adherent)* |
| #39 | *"The question is, can I achieve this goal? I hope so, but logically, glucose levels for people with diabetes are always higher than normal people. To plan something, it's easy for me, but to act according to my plan, that's where I face many challenges." (male, 24, non-adherent)* |
| #40 | *"I easily deviate from my plans, like if I go out with my friends, we won't go to have healthy food at a restaurant, of course, so I will have to eat unhealthy food like them. Also at home, I have to eat whatever is available, I don't have my own food. As I said, if I have supportive people surrounding me, the desire will come too. I mean having people there for me to support me." (male, 22, non-adherent)* |
| #41 | *"It has been very rare that I walk or go to the gym. I don't pay attention to making healthy food choices when I eat. I just eat randomly, everything that I find in front of me. For example, I take my breakfast, lunch, and dinner without any specific meal plan. Two years ago, I planned to lose weight, after I gave birth. I gained 10 kilograms while I was pregnant and I want to lose this extra weight, but I'm not able to do so until now. My target goal is 'for my HbA1c to reach 7 or less if it's possible, and I know it's my fault because I need to control my eating behaviors and do physical activity." (female, 23, non-adherent)* |
| #42 | *"Depending on what I know from my doctor, insulin always causes cells to store fat, which leads directly to weight gain, so my plan is to decrease my insulin dose intake through being physically active, and as a result, I eat small quantities." (female, 21, adherent)* |
| #43 | *"I already cut down on fast food, for the past two years I used to eat fast food 14 times a week. Yes, now and for the past 5 months. I eat fast food 3 times a week, so it's a miracle for me." (male, 19, non-adherent)* |
| #44 | *"Ya, I go to gatherings sometimes, I have an alternative plan, before I used to eat the whole plate, because I felt bad that it will be thrown in the rubbish, so I had to eat it all. But now I stopped doing that, I just eat like the size of my fist with plenty of fruits, vegetables, grilled food, and water." (female, 17, adherent)* |
| #45 | *"During summer breaks I register in gym, but when the university starts, I stop going, because I am too busy. Even I do not have plans to exercise at home, because I am too busy with studying and exams." (female, 20, non-adherent)* |
| #46 | *"I have a plan to do PA in late evenings during hot summer months, because it's too hot during the day to do activities outside the home. In winter I always exercise in the early mornings. Also, if it is too hot, I have a plan to go swimming instead of walking outside." (male, 24, adherent)* |
| #47 | *"I don't rely only on the information provided during my hospital's appointments, because they are 4 months apart. I look at stuff on the internet as well. Common information across more than one website gives me some insight if the info is correct. For example, if I'm going to eat something new for the first time, I attempt to know the number of carbs online but I try and I don't find it sometimes. I feel the search in English is more useful. There are more resources in English language." (female,19, non-adherent)* |
| #48 | *"Sometimes I receive information from my friends. It's rare to find a friend with diabetes. Nowadays everyone knows about diabetes so I can receive the information from anywhere. My friends get their info from WhatsApp or websites. For me, I rarely use the internet to get information." (male, 20, non-adherent)* |

*(Continued)*

**Table 5.** (Continued)

| Quote number | Quote and respondent |
|---|---|
| #49 | *"I rarely have hypos. But, in the past ten days, I had one around 6–7 times. Yes, I usually go to the gym after eating and I should cut down my dose by one third if I'm going to go to the gym after an hour of eating. Before starting to have hypos, I didn't know that I had to decrease it. It needs to be more specific in the way that health professionals deliver the needed information for patients. When I see my doctor for routine check-ups I would prefer if she /he sit with me for a time and discuss all I need to do in detail, including the specific types of exercises that suit me." (male, 17, adherent).* |
| #50 | *"The trainer at the gym kept pushing me to do more and more exercise, I had a hypo episode. I think trainers at gyms should be targeted in the education related to exercise and diabetes (male, 17, adherent).* |

## Discussion

The current study aimed to identify adherence towards HE and PA and the determinants of HE and PA in AYAs with T1D in Qatar. This study supports previous findings which showed that adherence to HE [95,96] and PA [73,96], was suboptimal. It is documented that non-adherence is a common problem in adolescents [97–99] and can continue until the mid-twenties [3,100]. Few people were adherent, and in general, those were the older participants. Previous data suggests that, compared to adolescents, young adults could be distinguished by cognitive capacity with more understanding of the consequences of actions [101] and more acceptance of their disease and self-care routine [102]. In the context of Qatar, the rapid socio-

**Table 6. Summary of findings.**

| Theme | Main outcome |
|---|---|
| **Pre-motivational factors** | |
| Awareness | The majority of participants were cognizant of their own behavior towards healthful eating (HE) and physical activity (PA). Some non-adherents overestimated their level of diabetes control and others were not aware of the need to adjust their behaviors. |
| Risk perception | The majority of both adherents and non-adherents recognized the susceptibilities of getting diabetes complications as a result of non-adherence to HE and PA as recommended. Yet this was not enough to promote adherence among non-adherents. Some participants did not link increased risk to low adherence to HE and PA. |
| **Motivational factors** | |
| Attitude: advantages | Irrespective of whether they were adherent or not, many respondents believed the advantages of HE and PA were linked to health benefits (both physical health and mental/psychological health) and more general advantages. Nevertheless, unlike adherents, non-adherents advantageous beliefs were not strong enough to bring them into action. |
| Attitude: disadvantages | Adherents mentioned the increased risk of injury and hypoglycemia being associated with adherence to PA and mentioned no disadvantages of HE. Non-adherents feared most PA-induced hypoglycemia. Dietary constraints and time consumption were mentioned as the main disadvantages to HE. |
| Self-efficacy | Non-adherents often encountered difficulties in adhering to HE and PA. |
| Social influence | Family impacted adherence mainly positively. Whereas peers impacted adherence negatively. Social environments have an important influence on adherence. |
| **Post-motivational factors** | |
| Action and coping planning | The majority of participants did not plan or execute relevant actions to realize the prescribed recommendations. They also reported sub-optimal goal setting, monitoring of self-care behavior and its outcomes, and maintaining behavior by resisting stimuli. |
| **Distal predisposing factors** | |
| Information factors | Information-seeking behavior varied among the participants. Mainly they sought information from health care providers and general internet websites. Specialized websites in the Arabic language are lacking. |

economic development and westernization of food habits played an important factor [56,59] and predicted unhealthy dietary habits among Qataris and residents [56]. Moreover, literature reviews from the Arab region revealed that physical inactivity was common among adults [103,104] and adolescents [104] general population. Some of the identified barriers were related to the specific cultural context of this region, such as the availability of and access to exercise facilities, hot weather, and lack of a social support system [103]. On the other hand, Ibrahim and colleagues (2018) found that sports facilities exist in most residential areas across Qatar [105]. However, there is a need for more data on the accessibility, utilization, and evaluation of sports facilities. Additionally, studies showed that Qatari students were less likely to be physically active than non-Qataris [59] and children with T1D were doing fewer outdoor activities [61]. The high prevalence of physical inactivity was associated with socio-economic factors and sedentary behaviors, such as the presence of housemaids, prolonged sitting at work or school, and extensive leisure time on screens (e.g., watching TV, using a computer, or playing video games) [59]. Cultural influences remain an area for future exploration in future research. A couple of participants stated that they adhered to HE but not to PA, or vice versa, because of personal beliefs about the expected outcome of behavior and due to self-efficacy factors. Usually, adherence across domains of diabetes self-management behaviors is not consistent [99,106]. Griva et al. (2000) [107] found that generalized and diabetes-specific self-efficacy among AYAs with T1D were correlated only to adherence to diet but not to PA [107]. Similarly, Mozzillo et al. (2017) [31] found that adherence to PA was lower in AYAs with T1D compared to adherence to diet, reflecting the influence of the disease on daily functioning [31]. Hence, understanding why patients choose to be adherent to PA and not HE or vice versa is needed.

## Pre-motivational factors: Awareness and risk perceptions

In this study, the majority of participants reported being cognizant of their own behaviors towards HE and PA. Yet, some non-adherents thought it was acceptable to consume extra-sugary foods as long as they increased their insulin dose. Research demonstrates that engaging in compensatory beliefs can result in maladaptive behavior without a feeling of guilt [108]. Additionally, increasing the daily insulin dose leads to an increase in weight, which further worsens insulin resistance [109,110]. There is some evidence that overweight and obesity are increasingly prevalent in people living with T1D [111,112] and that the focus of patients and HCPs on carbohydrate intake prevailed over the attention paid to the overall HE [22,23,113]. Hence, it is essential to stress the importance of HE to maintain a healthy weight and reduce cardiovascular risks.

Some non-adherent respondents either overestimated their level of diabetes control or were not aware of the need to adjust their behaviors. Thus, they assumed that they were doing well in terms of the level of HbA1c they had achieved, and this did not prompt them to eat healthy or to engage in PA. This correlates with a prior study which found that AYAs with T1D lacked an understanding of the meaning and the implications of the HbA1c test [114] and with previous studies which validated cognition of the need to change behavior as an important predictor of HE [115] and PA behaviors [116]. Quite a few non-adherents reported that they were either intending to change their HE or level of PA or they were in the preparation stage, but there was no committed effort. Previous data showed that people living with T1D in the contemplation and preparation stages were less likely to follow a healthy lifestyle and dietary habits [115,117]. Further, subjects who are in the contemplation stage will not be ready to cope with the disadvantages of new behavior. Therefore, they are more likely to relapse or discontinue [118]. Hence, they may require greater support to realize their actual level of diabetes control and their readiness for or stages of change.

In line with a previous study [119], we found that some participants did not adhere to HE and PA despite their foreknowledge about the risks associated with non-adherence. Joining their peers in social activities and maintaining their social image took priority over controlling their diabetes. While it is expected that higher risk perception should result in higher levels of adherence [120], some studies found that diabetes complications risks not only did not motivate adolescents with T1D to adhere, but were negatively related to adherence, due to low self-efficacy levels [121,122]. Plotnikoff et al. (2010) [123] found that coping appraisal variables (self-efficacy and response efficacy) were stronger predictors of intention to perform PA in people living with T1D compared to threat appraisal variables (perceived vulnerability and severity) [123]. We also found that some non-adherents did not link susceptibility to or severity of risks of complications to non-adherence. Some studies suggested that patients may continue unhealthy routines due to feelings that bad things will not happen to them [124] or not being convinced of immediate impacts on their health [125]. Previous results showed that AYAs with T1D engage in other risky behaviors, such as insulin and blood glucose monitoring non-adherence [3,126], alcohol use, illicit drug use, smoking, unprotected sex, and disordered eating behaviors [3,127]. Jaser et al. (2011) [127] reported that AYAs often have a lack of understanding and/or misunderstanding that these behaviors may risk their diabetes and health [127]. Beliefs about treatment effectiveness to control diabetes, treatment effectiveness to prevent complications, the perceived consequences and seriousness of diabetes were predictive of better dietary and PA self-management in adults with diabetes [128]. However, studies that examined personal models of diabetes in adolescents with diabetes showed varied results [128–130]. A study showed that the greater AYAs perceived their diabetes to be serious, the poorer their dietary self-care behaviors were (PA was not included in the analysis) [129]. Skinner et al., (2001) [130] found that the beliefs about treatment effectiveness to control diabetes has predicted better dietary but not PA self-care behaviors [130]. Neither beliefs about the seriousness of diabetes nor the treatment effectiveness to prevent long-term complications were predictive of HE or PA [130]. In another study, beliefs about treatment effectiveness to control diabetes and treatment effectiveness to prevent complications predicted better HE and PA behaviors, but the perceived threat of diabetes predicted poorer HE and PA [128]. Therefore, comprehensive education on the risks of non-adherence should be tailored to individual patients.

## Motivational factors: Attitude, social support, and self-efficacy

In the current study, non-adherents' advantageous beliefs, unlike adherents', were not enough to bring them into action. In contrast, some studies found that a more positive attitude improved adherence [131–133], while the perceived disadvantages of HE and PA were a major factor affecting non-adherence [134,135]. Three potential explanations could be that: (1) other factors such as low levels of self-efficacy and/or negative social support influenced suboptimal adherence; (2) disadvantages outweigh advantages; according to some participants; failing to see immediate effects of HE and PA on diabetes control has led to discontinuation of healthy behavior. It is known that people living with diabetes may be more likely to change their beliefs and behaviors if they can see how their existing practices lead to healthy outcomes [125,136]; and (3) perhaps the affective component of attitude (emotions created by the prospect of performing a behavior, e.g., feeling deprived of food, or fear of PA-induced hypoglycemia and fear of injuries were more influential on intention to perform behavior than the instrumental component of attitude (cognitive consideration of how advantageous performing a behavior would be). Earlier studies found that respondents with dietary constraints [137,138] and with FoH [139,140] reported not adhering to HE and PA, respectively. Thus, encouraging AYAs

with T1D to follow the recommendations of the State of Qatar National Physical Activity Guidelines (NPAG-Q) to seek specialized medical consultation and evaluation before exercising to determine the appropriate progress in the duration and intensity of PA, pre- and post-exercise meals is important to avoid hypoglycemia and injuries [141]. Past observations by other researchers [142,143] reported that measures of affective attitude were more predictive of intention than instrumental attitude. Clearly, further research is needed to examine the influence of attitude on behavior.

Regarding social influences, many non-adherent adolescents reported friends and family as mostly negatively affecting adherence, whereas adherents reported that having good health care and social support systems promoted adherence. Several studies demonstrated similar negative outcomes in social gatherings with peers [119,144]. On the other hand, a systemic review showed that peer involvement improved problem-solving and coping skills among people living with T1D [145]. Research has emphasized that family meal planning and gathering [146,147] and active family participation in PA [148,149] have improved adherence. However, other studies [98,150] found that family conflicts negatively impacted adherence. Thus, families' education and engagement to support a successful transition of self-management to AYAs and to avoid conflicts is needed. Also, the presence of a well-prepared health care support system increased trust in a provider's tailored advice and engagement in regular daily HE [151,152] and PA [2,153,154]. Our results indicated that exclusion from school activities has aggravated the emotional and social impact of diabetes, which confirms earlier results [155,156]. The International Society for Pediatric and Adolescent Diabetes (ISPAD) holds the position that adolescents must be able to manage their diabetes in the school setting without being excluded or discriminated [157]. Hence, educating the social environment to create a more supportive atmosphere for people living with diabetes should be enhanced.

Mirroring prior research, the results revealed a range of difficult situations, e.g., eating outside the home [38,158,159] or being busy [132,160–162]. Previous research conducted in Qatar demonstrated that the rapid socio-economic development and westernization of food habits also played an important factor in promoting unhealthy eating habits and a sedentary lifestyle [56,59]. Some difficult PA situations were related to the overwhelming burden of balancing insulin, diet, and PA before, during, and after PA, [26,160,163] and to the specific cultural context of this region, such as access to exercise facilities and a hot climate. In Qatar, despite the promising initiatives to promote PA at the national level, like formulating policies and organizing public sports activities such as "National Sports Day" and sports training at federation clubs [103,105], the prevalence of physical inactivity is high among the general youth population [103–105]. On the other hand, research has proven that people with T1D with high self-efficacy can motivate themselves [159,164], both directly through efficacy expectations and indirectly through perceived barriers [159], to make healthy food choices [164] and to incorporate PA into their daily routine [149,164,165]. Therefore, strengthening self-efficacy is a prerequisite for improving HE and PA adherence among this age group. Overall, given that motivation and intention are the immediate determinants of action [48] and increased patient motivation has been related to improving HE [44] and PA [166] adherence, it is crucial to assess what motivates AYAs with T1D to adhere to HE and PA recommendations.

### Post-motivational factors: Action and coping planning

Our results suggest that action and coping planning were lacking. Many non-adherents reflected on their planning as "implementation intentions" (such as "I plan to join the gym in the summer") rather than specified action planning. Detailed action planning should describe

more than a mere behavioral intention [167,168]. Literature has shown that action planning [48,169] and coping planning [167,170] can be an effective technique to prevent relapse. Ara-újo-Soares et al. found that action planning and coping planning were predictive of changes in PA in a sample of healthy adolescents [167]. Rohani, et al. (2018) [171] found that action planning and coping planning predicted HE behavior among adults with type 2 diabetes [171]. However, less evidence is available on action and coping planning effects on HE and PA in people living with T1D [172], thus requiring future research. Non-adherent respondents reported inadequate abilities to maintain their efforts towards adherence to PA or HE for a longer period. This could be attributed to the observed deficiencies in coping planning.

## Distal factors: Information factors

Regarding information factors, all participants wanted information and diabetes self-manage-ment education (DSME) to be tailored to their circumstances and to be continuous. Focus-group interviews by Litchfield and colleagues [154] indicated that the barriers to PA were related to the level of education they got from HCPs. Some non-adherent participants indi-cated having received contradictory messages regarding PA and HE. In a systematic review, some newly diagnosed T1D patients reported being advised by their HCPs not to exercise [173]. Therefore, it is important to coordinate the messages coming from a variety of sources. Many participants used the internet to look for information on HE and PA. However, these sites were not specific to diabetes and not targeted at AYAs. It is recognized that AYAs tend to use websites and other online resources to find information on diabetes management [12]. A noteworthy finding from the current study was that participants sometimes sought informa-tion from people without diabetes. A systematic review [174] revealed that friends and relatives were used as sources of information. Thus, information-seeking behavior from reliable sources should be fostered in AYAs with T1D.

Overall, this study has identified some factors that are known to influence behaviors involved in diabetes management. Furthermore, it has also highlighted determinants in the post-motivational (action planning and coping planning) which are limitedly investigated in T1D. The beneficial effects of these factors in increasing the likelihood of transition of inten-tions into actions have been demonstrated in various health conditions [170] and in type 2 dia-betes [175]. Therefore, more research is required to gain further insight into these factors in T1D to optimize adherence and improve diabetes outcomes. Additionally, it has drawn atten-tion to the needs of Arab patients with diabetes to have reliable educational material and resources in their native language.

## Strengths and limitations

This study has some strengths and limitations. First, it has added to our in-depth understand-ing of the determinants of adherence to HE and PA in young people living with T1D in Qatar, owing to the specificity and depth of the subjective information generated. Second, to mini-mize researcher bias, all interviews were conducted by the same researcher, who did not have a relationship with the participants. Third, considering the large volume of information gener-ated, adopting a framework analysis approach offered a systematic structure to easily manage, analyze, and identify themes [87].

This study also has some limitations. We recruited 20 participants based on some specific criteria for sample adequacy and verification of richness and saturation. However, it is still conceivable that our sample may not include certain categories. For instance, the views of young people who did not attend the follow-up appointments or were unwilling to participate were missed. These patients may have certain personality traits and views and are an important

target for further research. A recent systematic review indicated that younger adults, those with dismissive behavior and preoccupied attachment styles or with anxiety and/or depression, and those who had not attended diabetes education were less likely to attend appointments [176]. Non-attendance was associated with higher HbA1c [176] and with lower adherence to a healthy lifestyle [139,177]. Research suggests that young patients who are less inclined to disclose information regarding their diabetes are likely to be less adherent to diabetes management tasks and have a higher HbA1c [178]. Adolescents tend not to be open to talking about diabetes-related issues because of fear of discrimination and embarrassment. This leads to them missing opportunities to seek help regarding management, implicating the need for a more person-centered approach in T1D education. Nevertheless, it is suggested that the proliferation of qualitative research is the best way to ensure representation, rather than specifying such representation in samples [179]. Second, since saturation was deemed to be achieved, findings from this study may be transferable to similar groups. However, a larger sample size in a quantitative approach is needed to confirm our findings and increase generalizability. Third, while this study has highlighted specific determinants of non-adherence to HE related to insulin pump systems (e.g., some patients found insulin pumps have facilitated the use of corrective dose and gave them more freedom to eat whatever they wanted), we cannot draw enough conclusions on the effect of different types of insulin delivery devices on socio-cognitive factors (e.g., attitude, self-efficacy, etc.), and hence adherence to HE and PA. It was noted previously that insulin pumps offer users the flexibility to adjust insulin basal rates and boluses around exercise [180,181], but whether this has facilitated adherence to PA remains unclear. Therefore, more research is needed in this area to draw further comparisons and conclusions.

## Conclusions and recommendations

The suboptimal adherence in AYAs to HE and PA requires more attention. Supportive actions are needed to encourage adherence to a healthy lifestyle to achieve benefits in terms of glycemic control and overall health outcomes, with a special focus on adolescents. Interventions are needed to foster motivation by addressing the relevant determinants to promote adherence to these two behaviors in AYAs with T1D. Such approaches targeting lifestyle modification using modern educational means are particularly important when adherence to HE and PA is pronouncedly low. This study has identified some salient factors for AYAs with diabetes, which can help HCPs identify patients who are most likely to not adhere to HE and PA. The findings encourage diabetes professionals to include friends, family members, and staff at schools and gyms in diabetes education around HE and PA. Additionally, to regularly review the awareness of AYAs with T1D about the risks of non-adherence and identify ways to increase this awareness in a non-threatening manner, review their abilities to make specific action plans to increase and be prepared to cope with challenging situations. Thus, to promote adherence to HE and PA.

## Acknowledgments

The authors are grateful to all the participants who participated in the interviews. The authors would like to thank Hamad Medical Cooperation for providing a grant and the diabetes physicians who helped with participant recruitment. We would also like to thank the diabetes educator, Kawsar Mohamud, for her support in conducting this project, and Entisar Omer, Eman Faisel, and Heba Abo Shahla for assisting in transcribing the interviews.

## Author Contributions

**Conceptualization:** Hanan AlBurno, Liesbeth Mercken, Hein de Vries, Francine Schneider.

**Data curation:** Hanan AlBurno, Francine Schneider.

**Formal analysis:** Hanan AlBurno, Liesbeth Mercken, Francine Schneider.

**Funding acquisition:** Hanan AlBurno.

**Methodology:** Hanan AlBurno, Liesbeth Mercken, Hein de Vries, Dabia Al Mohannadi, Francine Schneider.

**Project administration:** Hanan AlBurno.

**Resources:** Dabia Al Mohannadi.

**Supervision:** Liesbeth Mercken, Hein de Vries, Francine Schneider.

**Validation:** Hein de Vries.

**Writing – original draft:** Hanan AlBurno.

**Writing – review & editing:** Liesbeth Mercken, Hein de Vries, Francine Schneider.

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
