## [Decision Letter · Decision Letter 0]

1 Mar 2022

PONE-D-21-16131Determinants of healthful eating and physical activity among adolescents and young adults with type 1 diabetes in Qatar: A qualitative studyPLOS ONE

Dear Dr. AlBurno,

Thank you for submitting your manuscript to PLOS ONE. After careful consideration, we feel that it has merit but does not fully meet PLOS ONE’s publication criteria as it currently stands. Therefore, we invite you to submit a revised version of the manuscript that addresses the points raised during the review process.

The manuscript has been evaluated by two reviewers, and their comments are available below.

The reviewers’ comments raise some overlapping concerns about additional detail and clarification of the methodology and statistical analyses, in addition to further depth of discussion in the Introduction and Discussion.

Could you please carefully revise the manuscript to address all comments raised?

We look forward to receiving your revised manuscript.

Kind regards,

Avanti Dey, PhD

Staff Editor

PLOS ONE

Journal Requirements:

4. We note you have included a table to which you do not refer in the text of your manuscript. Please ensure that you refer to Table 3 in your text; if accepted, production will need this reference to link the reader to the Table.

Reviewers' comments:

Reviewer's Responses to Questions

**Comments to the Author**

1. Is the manuscript technically sound, and do the data support the conclusions?

Reviewer #1: Yes

Reviewer #2: Yes

2. Has the statistical analysis been performed appropriately and rigorously? 

Reviewer #1: Yes

Reviewer #2: Yes

3. Have the authors made all data underlying the findings in their manuscript fully available?

Reviewer #1: Yes

Reviewer #2: Yes

4. Is the manuscript presented in an intelligible fashion and written in standard English?

Reviewer #1: Yes

Reviewer #2: Yes

5. Review Comments to the Author

Reviewer #1: This is a well written article which I enjoyed reading. In many ways the experiences of those living with type 1 diabetes in Qatar are similar to those in my own country of the UK. The heat is probably not a limit of physical activity however!

There are some things that could improve the article.

(1) Excepting direct quotes, please do not use the term diabetics (a disease condition should not be used to define a person). Please amend to people living with diabetes throughout. As the title explains that this work is in Type 1 diabetes, 'people living with diabetes should suffice.

(2) In the table of quotes, please define by age and sex but also by adherent, non adherent - it is vaguely irritating to have to go back to quote to identify status

(3) I am not quite certain I know how adherent and non-adherent was defined ? Was this by physician referral in purposive sampling, if so what criteria were used? Or was it by direct questioning initially in interview -again by what criteria?

(4) In insulin administration technique (table 2) - just use injectable pen (not disposable pen - quite a few insulins do not throw away pen, they replace cartridge)

(5) Could the English be reviewed please using a English language review tool? An example being line 449 Quite few non-adherent, should be Quite a few non-adherent. There are a few other examples but overall the English used in text is of very high quality.

(6) I am struck by Healthy Eating, do young people living with diabetes in Qatar use carbohydrate counting as technique. Can this be clarified please as it is the basis of dietary management in many countries now for diabetes with healthy eating on top of that, all guided by dietician sessions.

(7) In the limitations: do the authors feel that only having two patients with optimal control had an impact, as that seems like purposive sampling did not recruit sufficient to me - if you have five adherent but only two with optimal control which isn't that tight - we set 48mmol/mol for that now, it seems they may not in fact be that adherent? Please comment further on this observation.

Reviewer #2: Thank you for the opportunity to review your manuscript. The exploration of determinants for healthy eating and physical activity for AYAs with T1D is an important area of study and will likely be of interest to readers. Although the qualitative design is a strength of the study, there are a few concerns that need to be addressed. I have outlined these below for each section of the paper.

Introduction:

1) Please elaborate a bit more on how this study will be different than what others have done before.

a. The authors make the claim that not much work has been done with AYA in Qatar.

b. This is important, but please elaborate on this argument. What benefits would it have? What differences might you expect? Are there any important cultural expectations that might drive findings to be different from other publications?

2) Likewise, strengthen the argument for why qualitative work needs to be done.

3) Lines 110-126: Provide more research evidence for the model as it is used in predicting diabetes self-care behaviors. The authors describe the model, but add more citations/research support for the successful use of it in diabetes behaviors.

Methods:

1) Lines 141-142: Please clarify why the requirement for participants to be receiving insulin therapy for at least 2 months was made. That is - please add support for how 2 months is a significant amount of time to establish self-care behavior routines.

2) Describe in more detail how reliability and validity are established in this form of qualitative data analysis and coding process.

3) Please provide more detail about how adherence was measured. Was is just adherence to PA and HE? Or adherence to all recommended self-care behaviors? How did you determine who was considered "adherent" or "non-adherent"?

Results:

1) Quotes should be shared throughout the text rather than in a table. It was cumbersome to go back and forth between the written text and the table to read about the evidence. It made the story you are telling feel a bit disjointed.

a. Please note that this might be a journal requirement and if it is, then please disregard.

2) Overall, I think more quotes are needed to support some of the general results you are making. These are the bread and butter of qualitative work and what make the work interesting to read. Readers want to hear the voice of the participants, please include more. Here are sections I have identified that could use supporting quotes:

a. Line 231 - provide quotes of those who reported being adherent to PA and HE.

b. Lines 247-249:

c. Lines 270-272

d. 317-318

e. 384-387

f. 410-414

3) Please describe how you handled double codes )if any)?

a. For example, the results of adherent parents findings PA easier when others went with (Lines 358-360) could also be coded as social influences (lines 290)?

4) Table 3 is a nice summary. Thank you for including it.

Discussion:

1) Please describe how this study contributes something beyond what others have done.

2) Since the authors identify a need for this study based on the population, please bring that back into the discussion. What was similar to previous work? Anything new or insightful for this particular population?

3) Limitations may also include the different kinds of therapy that AYA with T1D use. It might be possible that participants who take shots have a different experience with HE and PA than those who are on an insulin pump system. Please include this in your limitations section.

The recommendations section should be expanded a bit. This is where the research gets to help others. Please provide a few more specific recommendations based on some of the general findings of your work (e.g., including friends in PA and HE management might boost adherence, finding ways to increase awareness of risk…).

6. PLOS authors have the option to publish the peer review history of their article (what does this mean?). If published, this will include your full peer review and any attached files.

Reviewer #1: No

Reviewer #2: No

---

## [Author Response · Author response to Decision Letter 0]

12 Apr 2022

Response to editor 

Dear editor,

Thank you for offering us an opportunity to revise our paper entitled “Determinants of healthful eating and physical activity among adolescents and young adults with type 1 diabetes in Qatar: a qualitative study” (PONE-D-21-16131). We herewith would like to resubmit our manuscript, in which we have addressed all issues raised by the editor and the two reviewers. 

In line with the suggestions from the editor and the two reviewers, we tried to provide more clarity and detail within our manuscript, by establishing a strong rationale for our study and additional, detailed information on the study design and in the discussion section. Furthermore, we removed retracted reference and replaced them with relevant current references and checked the manuscript for grammatical errors. 

We hope that you will consider the revised manuscript for publication in your journal.

Yours sincerely,

Hanan AlBurno

Liesbeth Mercken

Hein de Vries

Dabia Al Mohannadi

Francine Schneider

Journal requirement

Response to comment 1: We have modified the manuscript to meet PLOS ONE's style requirements.

Response to comment 2: Thank you for your comment. In our study, the principle investigator obtained the informed written consent/assent forms. We have further clarified this to indicate that the at the beginning of each interview, the PI obtained the informed written consent forms from young adults aged 18 years or above and the written assent forms from those under 18 years old with their carers’ consents, using the procedure approved by the ethics committee in the “Methods section: “Participants and recruitment” (lines 219-222). 

Response to comment 3: Thank you for noticing this error. We will correct the “Data availability statement” to indicate that participants’ quotes were shared anonymously within the body of the manuscript only, as to maintain the participants’ confidentiality. More data cannot be shared publicly to avoid violating the agreement to which the participants consented. Here is the text:

Data Availability: “The authors confirm that all data underlying the findings are fully available without restriction. All relevant data are within the manuscript” 

4. We note you have included a table to which you do not refer in the text of your manuscript. Please ensure that you refer to Table 3 in your text; if accepted, production will need this reference to link the reader to the Table.

Response to comment 4: Thank you, we made sure that all tables are cited including table 3. 

Response to comment 5: Using the updated reference list, we corrected references 8, 44, 115, 122, 124, 137, 141, 151, and 169. We also removed retracted references 3, 110, and 174, and replaced them with relevant current references. 

We removed reference number 151 from the submitted reference list (duplicate with reference number 126) and reference number 132 (duplicate with reference number 36). 

We changed the order of references 136, 137, 139, 140, and 141 in the submitted reference list to become 56, 59, 103,105, and 104 respectively in the updated reference list. The reason for this change is to match citations with the added paragraphs as suggested by the reviewers.

We added references 32, 33, 51-55, 57, 58, 60, 61, 64-66, 68, 89, 90, 93, 126-130, 175, and 179 -181 to cite the more requested details in the introduction, methods, results and discussion sections. 

Response to reviewers

We would like to thank the reviewers for their thorough reviews and constructive remarks and suggestions. We feel that the quality of this manuscript has been improved, due to the changes that were suggested. In the following we will explain how we incorporated these suggestions and solved problems indicated by the reviewers. The remarks of the reviewers are numbered and depicted first, followed by our responses to these comments. We hope that we have adequately addressed all comments and that the editor and reviewers deem the current adaptations appropriate and find the manuscript suitable for publication in the Journal of PloS One.

Hanan AlBurno

Liesbeth Mercken

Hein de Vries

Dabia Al Mohannadi

Francine Schneider

Response to reviewer #1

General Comment:

This is a well written article which I enjoyed reading. In many ways the experiences of those living with type 1 diabetes in Qatar are similar to those in my own country of the UK. The heat is probably not a limit of physical activity however!

Response to general comment: Thank you for the compliments. We are glad that the reviewer found the study well written and relevant.

1. Excepting direct quotes, please do not use the term diabetics (a disease condition should not be used to define a person). Please amend to people living with diabetes throughout. As the title explains that this work is in Type 1 diabetes, 'people living with diabetes should suffice.

Response to comment 1: We indeed agree and thank the reviewer for this valuable comment. We changed ‘diabetic” to “people living with diabetes”. Also, to be consistent in using terms, we replaced “patients” with “participants” where relevant. 

2. In the table of quotes, please define by age and sex but also by adherent, non adherent - it is vaguely irritating to have to go back to quote to identify status.

Response to comment 2: We added adherent, non-adherent to the respondents’ quotes in tables 3, 4, and 5 (lines 392, 555, and 661 respectively). 

3. I am not quite certain I know how adherent and non-adherent was defined ? Was this by physician referral in purposive sampling, if so what criteria were used? Or was it by direct questioning initially in interview -again by what criteria?

Response to comment 3: Adherence and non-adherence were self-reported and assessed in the interview. At the beginning of the interviews, the participants were familiarized with their diabetes care team and ADA recommendations for HE and PA, then based on their answers during the interview they were considered adherents, those respondents who reported that they always or most of the time follow the agreed recommendations from their diabetes care team for HE and PA, otherwise they were considered non-adherents.

We have indicated that the participants were familiarized with ADA recommendations for PA under the subheading “Interview process and procedure”. However, we rephrased this to: “At the beginning of each interview, the participants were familiarized with their diabetes care team and ADA recommendations for HE and PA” (line 253-254).

We also added “based on the agreed recommendations from their diabetes care team in table 1 (line 281). 

We further described the criteria for participants’ classification as adherents and non-adherents in “Data analysis section” (lines 279-284). 

4. In insulin administration technique (table 2) - just use injectable pen (not disposable pen - quite a few insulins do not throw away pen, they replace cartridge).

Response to comment 4: We changed disposable pen to injectable pen.

5. Could the English be reviewed please using a English language review tool? An example being line 449 Quite few non-adherent, should be Quite a few non-adherent. There are a few other examples but overall the English used in text is of very high quality.

Response to comment 5: We used an English language review tool to make sure that English language is reviewed and corrected. 

6. I am struck by Healthy Eating, do young people living with diabetes in Qatar use carbohydrate counting as technique. Can this be clarified please as it is the basis of dietary management in many countries now for diabetes with healthy eating on top of that, all guided by dietician sessions.

Response to comment 6: Yes, people living with diabetes in Qatar use carbohydrate counting as a technique to calculate insulin doses. In order to clarify this we, have added this information in the introduction section (see lines 97-99). Secondly, this was also stressed in the results section (lines 479-481) and in participant’s quote #26.

7. In the limitations: do the authors feel that only having two patients with optimal control had an impact, as that seems like purposive sampling did not recruit sufficient to me - if you have five adherent but only two with optimal control which isn't that tight - we set 48mmol/mol for that now, it seems they may not in fact be that adherent? Please comment further on this observation. 

Response to comment 7: Thank you for mentioning this point. We do not feel that only having two patients with optimal control had impacted the results. In this study we did not use HbA1c as an indicator of adherence to HE and PA, because it could be affected by adherence to insulin or other factors (such as hormonal changes, stress, ..etc). HbA1c were collected to reflect on participants’ awareness on their level of control and to explore whether they perceive adherence to HE and PA to influence diabetes outcome. Although, there was less representation of patients with optimal diabetes control, it is suggested that proliferation of a qualitative research is the best way to ensure representation, rather than specifying such representation in samples (Allmark, 2004).

In order to clarify this, we have added this information in the data analysis section (lines 286- 288). 

Response to reviewer #2

General Comment:

Thank you for the opportunity to review your manuscript. The exploration of determinants for healthy eating and physical activity for AYAs with T1D is an important area of study and will likely be of interest to readers. Although the qualitative design is a strength of the study, there are a few concerns that need to be addressed. I have outlined these below for each section of the paper.

Response to general comment: Thank you for the compliments, comments and suggestions. We are glad that the reviewer found the study subject important and that the qualitative design is a strength. 

Introduction:

1. Please elaborate a bit more on how this study will be different than what others have done before.

a. The authors make the claim that not much work has been done with AYA in Qatar.

b. This is important, but please elaborate on this argument. What benefits would it have? What differences might you expect? Are there any important cultural expectations that might drive findings to be different from other publications?

Response to comment 1: Thank you for your comment. In the introduction section, we have addressed the importance of this study compared to previously published articles with relevance to the selected age group, the comprehensive model used to predict distal and proximal factors affecting behavior, and the importance of exploring adherence behaviors and their determinants in different family, social, and cultural dynamics . However, we elaborated more on the specific socio-cultural context of this region which were identified in populations other than AYAs with T1D. 

We included more justification on the importance of the study compared to previously published literature, in the context of the social and cultural factors within Qatar and Middle East region in the last paragraph in the introduction section (lines 153-159). 

2. Likewise, strengthen the argument for why qualitative work needs to be done.

Response to comment 2: We indeed agree with the reviewer on the need to do qualitative research to understand patients’ perspectives. We included the rationale for adopting qualitative description design in the methods section (lines 182-186).

3. Lines 110-126: Provide more research evidence for the model as it is used in predicting diabetes self-care behaviors. The authors describe the model, but add more citations/research support for the successful use of it in diabetes behaviors.

Response to comment 3: We added more research evidence on the successful application of ICM in predicting and changing lifestyle behaviors in people with and without diabetes (lines 142-144). 

Methods:

1. Lines 141-142: Please clarify why the requirement for participants to be receiving insulin therapy for at least 2 months was made. That is - please add support for how 2 months is a significant amount of time to establish self-care behavior routines.

Response to comment 1: This was based on evidence from behavior psychology research which estimates the average time required for a new behavior (assuming that the patients will start the new healthy behaviors at the time of their diagnosis, although some may be already engaged in HE and PA before their diagnosis) to become automatic to be around 66 days. 

We rephrased “who have been receiving insulin therapy for at least 2 months “ to “who were diagnosed with T1D for at least 2 months” (to simplify sentence for the reader). We also added the rationale for recruiting patients who were diagnosed with diabetes for at least 2 months prior to the study to methods section: participants and recruitment (lines 198-202). 

2. Describe in more detail how reliability and validity are established in this form of qualitative data analysis and coding process.

Response to comment 2: Thank you for your comment. We indeed agree on the importance of establishing reliability and validity of data during coding and analysis processes. To ensure rigor during the coding process, the principal investigator coded all transcripts by allocating text segments to multiple codes in order to account for the complexity of data and double-checked against the codes, with two researchers reviewing the codes to enhance the quality of coding process. We provided more details on the codes were reviewed by two researchers in the “Qualitative data analysis” section (lines 297-299). 

Additionally, we have employed the verification strategies described by Morse et al., 2002 to ensure rigor. This was achieved by (1) employing inductive and deductive methods, the researchers (HB, LM and FS) regularly and iteratively discussed the coding system and analysis to validate the consistency in the application of codes, data interpretation, and formulation of findings, (2) we used a common conceptual framework with a priory defined codes, which were specific to particular interview questions, (3) we performed concurrent data collection and analysis to ensure methodological coherence, and (4) we checked for sampling adequacy and saturation. Moreover, in order to maintain consistency with the study aim, we applied an “ad hoc unitization strategy” (O’Connor et al, 2020) by including theoretically relevant simultaneous and interpretative codes in the coding frame and in the analysis and interpretation of data. We provided more details on this in “Qualitative data analysis” section (lines 314-323). 

3. Please provide more detail about how adherence was measured. Was is just adherence to PA and HE? Or adherence to all recommended self-care behaviors? How did you determine who was considered "adherent" or "non-adherent"?

Response to comment 3: Thank you for your question. In this study we focused only on adherence to HE and PA. Adherence and non-adherence was self-reported and assessed in the interview. At the beginning of interviews, participants were familiarized with their diabetes care team and ADA recommendations for HE and PA, then based on their answers during the interview they were considered adherents, those respondents who reported that they always or most of the time follow the agreed recommendations from their diabetes care team for HE and PA, otherwise they were considered non-adherents.

We have indicated that the participants were familiarized with ADA recommendations for PA under the subheading “Interview process and procedure”. However, we rephrased this to: “At the beginning of each interview, the participants were familiarized with their diabetes care team and ADA recommendations for HE and PA” (line 253-254).

We also added “based on the agreed recommendations from their diabetes care team in Table 1 (line 281). 

We further described the criteria for participants’ classification as adherents and non-adherents in “Data analysis section” (lines 279-284). 

Results:

1. Quotes should be shared throughout the text rather than in a table. It was cumbersome to go back and forth between the written text and the table to read about the evidence. It made the story you are telling feel a bit disjointed.

a. Please note that this might be a journal requirement and if it is, then please disregard.

Response to comment 1: Thank you for your suggestion. We understand that including the quotes throughout the text may be more convenient for the reader. However, considering the relatively long manuscript and rich information, we tried our best to not over-length it, to ensure that the readers will go through it. Therefore, we decided for ‘in table quotes’. However, we would like to offer more clarity to the reviewer and have therefore opted for an alternative solution that can hopefully meet the reviewer’s need, while still fitting with our aims of the table by separating ‘Table 3. Interviewee quotes’ into 3 tables. Table 3. Interviewee quotes - pre-motivational factors’, ‘Table 4. Interviewee quotes - motivational factors’, and ‘Table 5. Post-motivational and distal information factor’ (lines 392, 555, and 661 respectively). 

2. Overall, I think more quotes are needed to support some of the general results you are making. These are the bread and butter of qualitative work and what make the work interesting to read. Readers want to hear the voice of the participants, please include more. Here are sections I have identified that could use supporting quotes:

a. Line 231 - provide quotes of those who reported being adherent to PA and HE.

b. Lines 247-249:

c. Lines 270-272

d. 317-318

e. 384-387

f. 410-414

Response to comment 2: Thank you for your comment. We added quotes relevant to the specified results.

a) [Quote # 1]: (adherent to both HE and PA). We also added [Quote # 2]: adherent to HE not to PA.

b) We added [Quote #6] to support misjudgement about the level of control and [Quote # 7], to support perception of no need to change behavior, because of wrong perception that HE and PA will not affect level of control. 

c) We added [Quote #13] to support advantages of overall adherence. 

d) We added [Quote #23]: to support AYAs’ perceptions about the negative social influence from HCPs and parents. 

e) We added [Quote #44] to support coping plans for HE, [Quote #45] to support lack of coping plans for PA, and [Quote #46] to support coping plans for PA.

f) We added [Quote #50] to support the need to educate trainers at gym. 

We also made a few additional changes in the quotes to fit the purpose of text and to support results.

a) we removed [Quote #9] , because quote #14 contained more description of advantages to PA.

b) added few sentences to quote #21 to support respondent’s feeling about the family interaction.

c) added few sentences to quote #26 to support respondent’s reporting difficulty in CHO counting.

d) changed order of quotes #27 and #28 to match text.

e) changed order of quotes #29 and #30 to match text.

f) added few sentences to quote #47 to support respondent’s view for source of information.

g) added few sentences to quote #49 to support the need for more information on what type of exercises are suitable for T1D and on adjusting insulin dose around PA in order to avoid hypoglycemia.

3. Please describe how you handled double codes )if any)?

a. For example, the results of adherent parents findings PA easier when others went with (Lines 358-360) could also be coded as social influences (lines 290)?

Response to comment 3: We used a common conceptual framework with a priory defined codes, which were specific to particular interview questions. Additionally, in order to maintain consistency with the study aim, we applied an “ad hoc unitization strategy” described by O’Connor et al, 2020, through including theoretically relevant simultaneous and interpretative codes in the coding frame and in analysis and interpretation of data. We have added more clarification in “Qualitative data analysis” section (lines 318-319) and lines (321-323).

4. Table 3 is a nice summary. Thank you for including it.

Response to comment 4: Thank you. We are glad that the reviewer found the summary table useful. 

Discussion:

1. Please describe how this study contributes something beyond what others have done.

Response to comment 1: Overall, this study has identified some factors which are known to influence behaviors involved in diabetes management. Furthermore, it has also highlighted determinants in the post-motivational (action planning and coping planning) which are limitedly investigated in T1D. The beneficial effects of these factors in increasing the likelihood of transition of intentions into actions have been demonstrated in various health conditions (Sniehotta et al., 2005) and in type 2 diabetes (Vluggen et al., 2018). Therefore, more research is required to gain further insight into these factors in T1D to optimize adherence and improve diabetes outcomes. Additionally it has drawn attention to the needs of Arab patients with T1D to have reliable educational material and resources in their native language.

We added the contribution provided by this research compared to previously published literature in the discussion section (lines 1029-1036). 

2. Since the authors identify a need for this study based on the population, please bring that back into the discussion. What was similar to previous work? Anything new or insightful for this particular population?

Response to comment 2: We compared the similarities and differences between our results and results from previously published articles on same subject throughout the discussion section. However, we provided more evidence related to AYAs with and without diabetes in Qatar and Middle East region (lines 688-701). Also, related to risk perception (lines 775-802) because conflicting results were published previously. 

3. Limitations may also include the different kinds of therapy that AYA with T1D use. It might be possible that participants who take shots have a different experience with HE and PA than those who are on an insulin pump system. Please include this in your limitations section. The recommendations section should be expanded a bit. This is where the research gets to help others. Please provide a few more specific recommendations based on some of the general findings of your work (e.g., including friends in PA and HE management might boost adherence, finding ways to increase awareness of risk…).

Response to comment 3: Thank you for comment. Although this study has highlighted specific determinants of non-adherence to HE related to insulin pump systems (e.g., some patients found insulin pumps have facilitated the use of corrective dose and give them more freedom to eat whatever they want. Yet, we cannot draw enough conclusions on the effect of different types of insulin delivery devices on socio-cognitive factors (e.g., attitude, self-efficacy. etc.), and hence adherence to HE and PA. It was noted previously that insulin pumps offer the users the flexibility to adjust insulin basal rates and boluses around exercise (Tagougui, et al., 2018; Colberg et al., 2021), but whether this facilitate adherence to PA remains unclear. Therefore, more research is needed in this area to draw further comparisons and conclusions. 

We added the limitation and draw the suggestion for the need of more research to identify the effect of different types of insulin delivery devices on adherence to HE and PA (lines 1071-1088). 

We also expanded the recommendation section as per your suggestion (lines 1096-1102). And here is the text: 

“This study has identified some salient factors for AYAs with diabetes, which can help HCPs to identify patients who are most likely to not adhere to HE and PA. The findings encourage diabetes professionals to include friends, family members and staff at schools and gyms in diabetes education around HE and PA. Additionally, to regularly review the awareness of AYAs with T1D about the risks of non-adherence and identify ways to increase this awareness in a non-threatening manner, also to review their abilities to make specific action plans to increase and be prepared to cope with challenging situations. Thus, to promote adherence to HE and PA.” 

The authors would like to thank the editor, the two reviewers for their dedicated time to review our manuscript and for their comments. 

References:

• Allmark P. Should research samples reflect the diversity of the population? J Med ethics. 2004;30(2): 185-9. doi:10.1136/jme.2003.004374.

• Colberg SR, Kannane J, Diawara N. Physical activity, dietary patterns, and glycemic management in active individuals with type 1 diabetes: an online survey. Int J Environ Res Public Health. 2021;18(17):9332-52. doi:10.3390/ijerph18179332

• Morse JM, Barrett M, Mayan M, Olson K, Spiers J. Verification strategies for establishing reliability and validity in qualitative research. Int J Qual Methods. 2002;1(2): 13-22. doi:10.1177/160940690200100202.

• O’Connor C, Joffe H. Intercoder reliability in qualitative research: debates and practical guidelines. Int J Qual Methods. 2020;19: 160940691989922-2. doi:10.1177/1609406919899220.

• Sniehotta F, Schwarzer R, Scholz U, Schuz B. Action planning and coping planning for long-term lifestyle change: theory and assessment. Eur J Soc Psychol. 2005; 35: 565-76. doi:10.1002/ejsp.258.

• Tagougui S, Taleb N, Rabasa-Lhoret R. The benefits and limits of technological advances in glucose management around physical activity in patients type 1 diabetes. Front Endocrinol. 2018;9: 818-8. doi:10.3389/fendo.2018.00818.

• Vluggen S, Hoving C, Schaper NC, de Vries H. Exploring beliefs on diabetes treatment adherence among Dutch type 2 diabetes patients and healthcare providers. Patient Edu Couns. 2018;101(1): 92-98. doi:10.1016/j.pec.2017.07.009.

---

## [Decision Letter · Decision Letter 1]

22 Jun 2022

Determinants of healthful eating and physical activity among adolescents and young adults with type 1 diabetes in Qatar: A qualitative study

PONE-D-21-16131R1

Dear Dr. Hanan AlBuno 

We’re pleased to inform you that your manuscript has been judged scientifically suitable for publication and will be formally accepted for publication once it meets all outstanding technical requirements.

Kind regards,

Enock Madalitso Chisati, PhD

Academic Editor

PLOS ONE

Additional Editor Comments (optional):

Reviewers' comments:

Reviewer's Responses to Questions

**Comments to the Author**

1. If the authors have adequately addressed your comments raised in a previous round of review and you feel that this manuscript is now acceptable for publication, you may indicate that here to bypass the “Comments to the Author” section, enter your conflict of interest statement in the “Confidential to Editor” section, and submit your "Accept" recommendation.

Reviewer #1: All comments have been addressed

Reviewer #2: All comments have been addressed

2. Is the manuscript technically sound, and do the data support the conclusions?

Reviewer #1: Yes

Reviewer #2: Yes

3. Has the statistical analysis been performed appropriately and rigorously? 

Reviewer #1: N/A

Reviewer #2: Yes

4. Have the authors made all data underlying the findings in their manuscript fully available?

Reviewer #1: Yes

Reviewer #2: Yes

5. Is the manuscript presented in an intelligible fashion and written in standard English?

Reviewer #1: Yes

Reviewer #2: Yes

6. Review Comments to the Author

Reviewer #1: Thank you for addressing my comments in first review. The text now makes for an interesting addition to the literature on diabetes adherence in young people.

Reviewer #2: My comments and issues with the manuscript have been adequately addressed. I do not have any further comments or concerns with the manuscript.

7. PLOS authors have the option to publish the peer review history of their article (what does this mean?). If published, this will include your full peer review and any attached files.

Reviewer #1: No

Reviewer #2: No

---

## [Editor Report · Acceptance letter]

27 Jun 2022

PONE-D-21-16131R1 

Determinants of  healthful eating and physical activity among adolescents and young adults with type 1 diabetes in Qatar: A qualitative study 

Dear Dr. AlBurno:

I'm pleased to inform you that your manuscript has been deemed suitable for publication in PLOS ONE. Congratulations! Your manuscript is now with our production department. 

Kind regards, 

on behalf of

Dr. Enock Madalitso Chisati 

Academic Editor

PLOS ONE